# Correlations in sleeping patterns and circadian preference between spouses

Rebecca C. Richmond [1,2 ✉], Laurence J. Howe [1,2], Karl Heilbron[3,4], Samuel Jones [5], Junxi Liu[1,2,6], 23andMe Research Team*, Xin Wang [7], Michael N. Weedon[8], Martin K. Rutter[9,10], Deborah A. Lawlor [1,2,11], George Davey Smith [1,2,11] & Céline Vetter[12]

Spouses may affect each other's sleeping behaviour. In 47,420 spouse-pairs from the UK Biobank, we found a weak positive phenotypic correlation between spouses for self-reported sleep duration (r = 0.11; 95% CI = 0.10, 0.12) and a weak inverse correlation for chronotype (diurnal preference) (r = −0.11; −0.12, −0.10), which replicated in up to 127,035 23andMe spouse-pairs. Using accelerometer data on 3454 UK Biobank spouse-pairs, the correlation for derived sleep duration was similar to self-report (r = 0.12; 0.09, 0.15). Timing of diurnal activity was positively correlated (r = 0.24; 0.21, 0.27) in contrast to the inverse correlation for chronotype. In Mendelian randomization analysis, positive effects of sleep duration (mean difference=0.13; 0.04, 0.23 SD per SD) and diurnal activity (0.49; 0.03, 0.94) were observed, as were inverse effects of chronotype (−0.15; −0.26, −0.04) and snoring (−0.15; −0.27, −0.04). Findings support the notion that an individual's sleep may impact that of their partner, promoting opportunities for sleep interventions at the family-level.

[1] Medical Research Council Integrative Epidemiology Unit, University of Bristol, Bristol, UK. [2] Population Health Sciences, Bristol Medical School, University of Bristol, Barley House, Oakfield Grove, Bristol, UK. [3] Department of Psychiatry and Psychotherapy, Charité Universitätsmedizin, Berlin, Germany. [4] Stanley Center for Psychiatric Research, Broad Institute of Harvard and MIT, Cambridge, MA, USA. [5] Institute for Molecular Medicine FIMM, HiLIFE, University of Helsinki, Helsinki, Finland. [6] Oxford Population Health, Nuffield Department of Population Health, University of Oxford, Oxford, UK. [7] 23andMe, Inc., 223 N Mathilda Avenue, Sunnyvale, CA, USA. [8] Genetics of Complex Traits, University of Exeter Medical School, Exeter, UK. [9] Division of Endocrinology, Diabetes & Gastroenterology, School of Medical Sciences, Faculty of Biology, Medicine and Health, University of Manchester, Manchester, UK. [10] Diabetes, Endocrinology and Metabolism Centre, Manchester University NHS Foundation Trust, NIHR Manchester Biomedical Research Centre, Manchester Academic Health Science Centre, Manchester, UK. [11] National Institute of Health Research Biomedical Research Centre, University of Bristol, Bristol, UK. [12] Circadian and Sleep Epidemiology Laboratory, Department of Integrative Physiology, University of Colorado Boulder, Boulder, CO, USA. *A list of authors and their affiliations appears at the end of the paper. ✉email: rebecca.richmond@bristol.ac.uk

nsufficient and disturbed sleep are pervasive features of society, with more than a quarter of US adults reporting sleeping six or fewer hours per night[1], and over a third of adults reporting insomnia[2]. In addition to insomnia and short sleep duration, symptoms of sleep disturbance include long sleep duration, difficulty waking up in the morning and daytime sleepiness[3]. Sleep problems have been strongly associated with occupational accidents[4] and loss of productivity[5], as well as elevated risk of cardiovascular disease[6], metabolic disease[7,8], depression[9] and some forms of cancer[10].

Sleep patterns vary across the life course[3], are affected by ageing processes[11] and have been associated with demographic and socioeconomic characteristics (e.g., marital status, employment and parenthood)[1]. Men and women exhibit differences in sleep-wake patterns which vary with age[3]. For example, on average, men have a more pronounced later chronotype (evening preference) than women, especially in early adulthood, but this observed difference diminishes over time[12].

Within co-habiting couples, it is of interest to investigate the interdependence in sleep patterns since this may exacerbate sleep problems, which could have social, psychological, and physical health implications[13]. Establishing whether the sleep habits of a spouse could serve as a risk factor for an individual's own poor sleep would enhance our understanding of the familial impacts on sleep and sleep-related ill health, and would promote opportunities for interventions aimed at the family level.

Within a couple, it is plausible that sleep traits are correlated. Spousal concordance is well established in humans for several characteristics[14], including cardiometabolic health[15], smoking[15], alcohol consumption[16], educational level[17], language and culture[18]. Spouses tend to be positively correlated for most measured phenotypes, and this may represent positive assortative mating or social homogamy (whereby individuals select phenotypically similar partners), interactions after partnership (where an individual's behaviour influences that of their spouse) which may result in convergence over time, or confounding by shared environmental factors[19].

In a study of 46 couples, actigraphy-assessed sleep movements were more frequent when couples were sleeping together vs. when sleeping apart (6% vs. 5.5% probability of movement onset per hour asleep), and yet subjective sleep was generally reported to be worse when sleeping apart (44% reported sleeping better with their partner present vs. 22% when their partner was away)[20]. The same study found that females reported being disturbed more often by their partner than was the case for males (9% vs. 6%). Another study of 36 couples evaluated the interdependence of sleeping patterns based on several actigraphy-assessed measures and found strong correlations in bedtime (intraclass correlation (ICC) = 0.42, $p < 0.01$), sleep latency (ICC = 0.25, $p < 0.001$), light/dark ratio (ICC = 0.28, $p < 0.001$) and wake bouts (ICC = 0.42, $p < 0.001$) between couples[21].

Previous studies using actigraphy measures to investigate spousal sleeping patterns have been limited in terms of sample size. While larger studies have investigated self-reported sleep traits among spouses[22,23], they may suffer from bias due to individuals' perception and recall of sleeping patterns, which may differ between men and women. Previous observational studies investigating both self-reported and objectively assessed sleeping patterns between spouses may also be biased by confounding (i.e., by shared socioeconomic and lifestyle factors) and it can be difficult to determine the directionality in correlated sleep patterns between spouses (i.e., the extent to which one spouse influences the sleep patterns of the other, and vice versa).

Mendelian randomisation (MR) is a method that uses genetic variants to evaluate causality between two traits by minimising the risk of confounding and reverse causation[24–26]. While it is typically used to investigate the effects of traits within the same individual, it may be extended to investigate the effect of one individual on another[27]. This builds on the concept of social or indirect genetic effects[28–31], where the genotype of an individual influences the phenotype of an individual's contacts (spouses, parents or friends). Evidence for indirect genetic effects between couples across a range of socioeconomic, lifestyle and behavioural phenotypes has been recently identified in large-scale population datasets[32,33]. One study used genetic data from co-habiting spouses in the UK Biobank to investigate the possible causes of spousal similarity for alcohol behaviour[34]. A similar approach can be used to investigate spousal correlations in sleep behaviour.

Accelerometer-based assessments of sleep patterns, which have been demonstrated to be correlated with gold-standard polysomnography data[35], are now available in much larger studies such as the UK Biobank[36]. The UK Biobank also has data on self-reported sleep traits, as well as genetic data, and contains ~50,000 co-habiting spouses[34].

In this study, we aimed to investigate similarities in sleeping patterns between spouses in UK Biobank and 23andMe, Inc. ($n = 174{,}455$ spouse-pairs). We evaluated five self-reported sleep and four accelerometer-based sleep measures in UK Biobank. Four of the self-reported sleep traits were also available for assessment in 23andMe. If similarities in sleeping patterns and circadian preference are observed, this may represent assortative mating by sleep traits, sleep interactions after partnership (where an individual's sleep pattern influences that of their spouse) or confounding by shared environmental factors (Fig. 1). To minimise the risk of confounding, we performed MR using genetic variants associated with the nine sleep traits in UK Biobank to estimate the effect of an individual's sleep patterns on those of their spouse. To determine whether any effects represent assortative mating, we also investigated genetic concordance for sleep traits between spouses, which would imply that an effect exists prior to pairing (since genotypes cannot be modified). Finally, we conducted sensitivity analyses to evaluate potential bias in the MR analysis.

## Results

**UK Biobank: sleep characteristics between spouse-pairs**. Of the 47,549 derived spouse-pairs in the UK Biobank, 47,420 (99.7%) had reported information about their sleep in a touchscreen questionnaire completed at baseline and 3,454 pairs (7.3%) had valid data from a triaxial accelerometer device (Axivity AX3) worn for a continuous period of up to 7 days between 2.8 and 8.7 years after study baseline, from which several sleep measures were derived (Supplementary Fig. 1 and Supplementary Note 1).

The mean age of female and male spouses at baseline was 56.8 (SD 7.3) and 58.5 (7.3) years, respectively. Reported sleep duration was similar between both females and males (mean (SD): 7.3 (1.1) and 7.2 (1.0) hours). Males were slightly more likely to report no chronotype preference (12.8% males vs. 8.4% females) and to report an extreme evening preference (7.0% males vs. 6.4% females), while females were slightly more likely to report an extreme morning preference (23.9% females vs. 22.6% males). Male spouses found waking up in the morning easier, with 40.2% finding it very easy compared with 27.4% of women. Female spouses reported more frequent insomnia symptoms, with 82.5% reporting that they sometimes or usually had symptoms, compared with 69.6% of males. Males were more likely to say that their spouse complained about their snoring (53.6% vs. 30.2%) (Table 1).

Of those spouse pairs who participated in the accelerometer assessment, the mean age of females and males was 63.1 (SD 6.9) and 64.8 (7.0) years when worn. Estimated nocturnal sleep duration was similar between male and female spouses (7.5 (0.8)

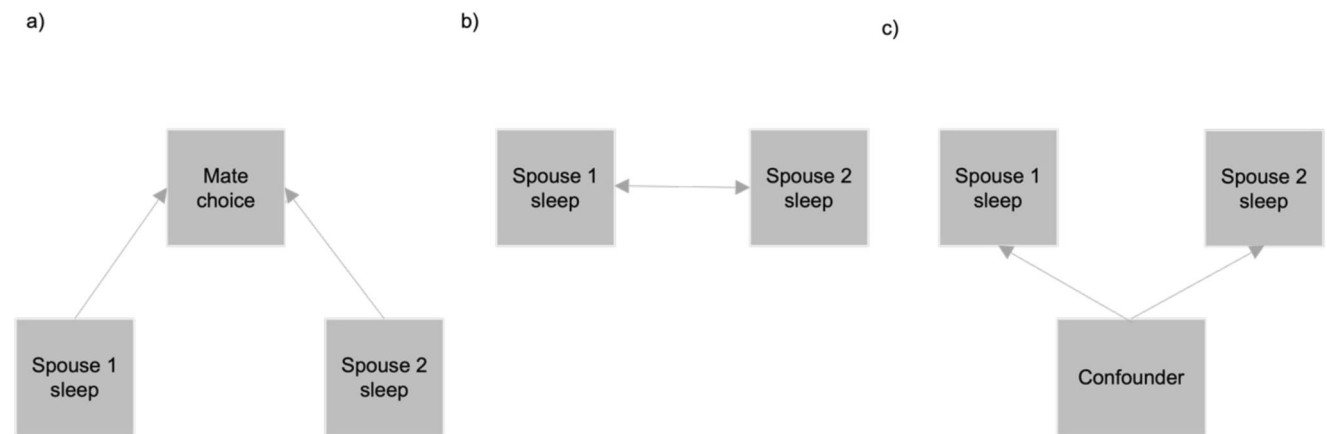

**Fig. 1 Scenarios for spousal concordance of sleep traits. a** Assortative mating—individuals are more likely to select a mate with similar sleeping behaviour.
**b** Partner interaction—after the partnership, spouses influence each other's sleeping behaviour. **c** Confounding—shared environmental factors influence the
sleeping behaviour of the spouses.

| Table 1 Sleep traits among male and female spouses in UK Biobank. | | | |
| --- | --- | --- | --- |
| | | **Female spouses** | **Male spouses** |
| **Self-reported traits** | **N (spouse-pairs)** | **Mean (SD)** | **Mean (SD)** |
| Age at baseline assessment | 47,420 | 56.8 (7.3) | 58.5 (7.3) |
| Sleep duration | 47,169 | 7.3 (1.1) | 7.2 (1.0) |
| | | **% (n)** | **% (n)** |
| Chronotype | 47,235 | | |
| Extreme evening preference | | 6.4 (3003) | 7.0 (3291) |
| Intermediate evening preference | | 26.2 (12,360) | 24.6 (11,629) |
| No preference | | 8.4 (3980) | 12.8 (6047) |
| Intermediate morning preference | | 35.2 (16,609) | 32.9 (15,558) |
| Extreme morning preference | | 23.9 (11,283) | 22.6 (10,710) |
| Ease of waking up | 47,325 | | |
| Not at all easy | | 4.2 (2007) | 1.9 (889) |
| Not very easy | | 15.7 (7416) | 9.3 (4400) |
| Fairly easy | | 52.7 (24,930) | 48.6 (23,010) |
| Very easy | | 27.4 (12,972) | 40.2 (19,026) |
| Insomnia symptoms frequency | 47,369 | | |
| Never/rarely | | 17.5 (8291) | 30.4 (14,393) |
| Sometimes | | 50.3 (23,828) | 46.0 (21,789) |
| Usually | | 32.2 (15,250) | 23.6 (11,187) |
| Snoring | | | |
| No | 45,546 | 69.8 (31,785) | 46.4 (21,135) |
| Yes | | 30.2 (13,761) | 53.6 (24,411) |
| **Accelerometer-derived traits** | | **Mean (SD)** | **Mean (SD)** |
| Age at accelerometer assessment | 3454 | 63.1 (6.9) | 64.8 (7.0) |
| L5-timing | 3454 | 27.3 (1.0) | 27.3 (1.0) |
| Sleep duration | 3454 | 7.5 (0.8) | 7.3 (0.9) |
| Nocturnal sleep episodes | 3454 | 16.9 (3.5) | 17.6 (3.8) |
| Sleep efficiency | 3454 | 0.78 (0.06) | 0.75 (0.07) |

and 7.3 (0.9) hours) and was consistent with the self-reported
estimates. Timing of the least-active 5 h the day (L5-timing) was
equivalent between males and females, with the mean midpoint
estimated as 3.18 am (SD 1.0). Despite reporting more frequent
insomnia and difficulty getting up in the mornings at baseline,
females who wore the accelerometer had more efficient sleep
(0.78 (0.06) vs. 0.75 (0.07)) with fewer sleep episodes (16.9 (3.5) vs.
17.6 (3.8)) than their male spouses (Table 1).

The UK Biobank spouse-pairs were slightly older (mean age
57.6 years (7.4)) than the remainder of the UK Biobank cohort
(56.3 years (8.2)), had a lower Townsend deprivation index
(mean −2.2 (2.5) vs. −1.1 (3.2)), were less likely to be current

smokers (7.1% vs. 11.4%) and more likely to abstain from alcohol
(22.3% vs. 19.9%). However, they were less likely to be employed
(54.3% vs. 58.3%) and to have a university degree (31.6% vs.
33.0%) (Supplementary Data 1). Those individuals in spouse-
pairs who participated in the accelerometer assessment had a
lower Townsend deprivation index (mean −2.4 (2.3) vs. −2.1
(2.5)) and were more likely to be employed (56.3%) and to have a
university degree (41.3%) than those spouse-pairs who did not
participate (53.8% and 29.0%). They also had a lower body mass
index (BMI), were less likely to be current smokers (4.8% vs.
7.7%) and more likely to abstain from alcohol (24.0% vs. 21.9%)
(Supplementary Data 2).

72.4% of the UK Biobank participants with genetic data reported living with a spouse. They were less likely to report having an extreme evening preference (7.2%) compared with those living with someone other than a spouse (9.9%) or living alone (10.7%); found it less difficult waking up in the morning (3.2% vs. 6.0% and 5.6%) and experienced less frequent insomnia symptoms (27.1% vs. 29.3% and 31.7%). They were more likely to report snoring (40.3% vs. 28.9% and 27.1%), which is likely an artefact of how this question was asked: "Does your partner or a close relative or friend complain about your snoring?". Both self-reported and accelerometer-derived sleep duration were similar between the household categories, as were accelerometer-derived diurnal activity (L5-timing), nocturnal sleep episodes and sleep efficiency. Findings were similar when stratified by sex, except for snoring behaviour which was reported at a similar prevalence among women in the different household categories (Supplementary Data 3). Sleep traits among participants who also had a spouse in the UK Biobank cohort (UK Biobank spouse-pairs) were generally similar to those who reported living with a spouse not in UK Biobank (Supplementary Data 1).

**23andMe: sleep characteristics between spouse-pairs.** The mean age of female and male spouses in 23andMe was 62.5 (SD 11.5) and 64.5 (11.7), respectively. Reported sleep duration was similar between both females and males, but less than that reported in UK Biobank (mean (SD): 5.9 (1.2) and 5.9 (1.1) hours). Unlike in UK Biobank, males in 23andMe were less likely to report having an evening preference than their spouses (34.5% vs. 41.7%), Female spouses were more likely to report having been diagnosed or treated with insomnia (20.0% vs. 11.9%). Male spouses were more likely to report that they snored (56.7% vs. 37.2%) (Supplementary Data 4).

**UK Biobank: spousal phenotypic correlation for sleep traits.** Self-reported and accelerometer-derived sleep traits were correlated between spouse pairs, except for insomnia and snoring. Weak positive correlations were found for L5-timing ($r = 0.24$; 95%CI = 0.21, 0.27), self-reported and accelerometer-derived sleep duration ($r = 0.11$; 0.10, 0.12 and $r = 0.12$; 0.09, 0.15, respectively), sleep efficiency ($r = 0.07$; 0.04, 0.10), number of sleep episodes ($r = 0.08$; 0.05, 0.11) and ease of waking ($r = 0.04$; 0.04, 0.05). An inverse correlation was observed for chronotype ($r = -0.11$; $-0.12$, $-0.10$) (Fig. 2). Phenotypic correlations were generally smaller in magnitude than other sociodemographic and lifestyle factors considered, which were all positively correlated between spouses ($r = 0.13$ to 0.47) (Fig. 2). Corresponding risk and mean differences obtained from multivariable (MV) regression were very similar to the phenotypic correlations (as expected given that MV regression of SD on SD ~ partial correlation) (Table 2). Weak cross-trait correlations were also evident between the spouses, with the largest positive correlation between snoring and insomnia ($r = 0.10$; 0.09, 0.11) and the largest inverse correlation between L5-timing (later diurnal activity) and chronotype (morning preference) ($r = -0.07$; $-0.09$, $-0.05$) (Supplementary Fig. 2).

**23andMe: spousal phenotypic correlation for sleep traits.** Self-reported sleep traits were also correlated between spouse-pairs in 23andMe (Supplementary Data 5). Similar to UK Biobank, sleep duration was positively correlated between spouses ($r = 0.12$; 0.09, 0.15) while chronotype was inversely correlated ($r = -0.13$; $-0.14$, $-0.12$) (Fig. 2). Weak positive correlations were also observed for insomnia ($r = 0.07$; 0.06, 0.07) and snoring ($r = 0.05$; 0.03, 0.07), which were larger in magnitude than in UK Biobank (Fig. 3). Again, weak cross-trait correlations were observed

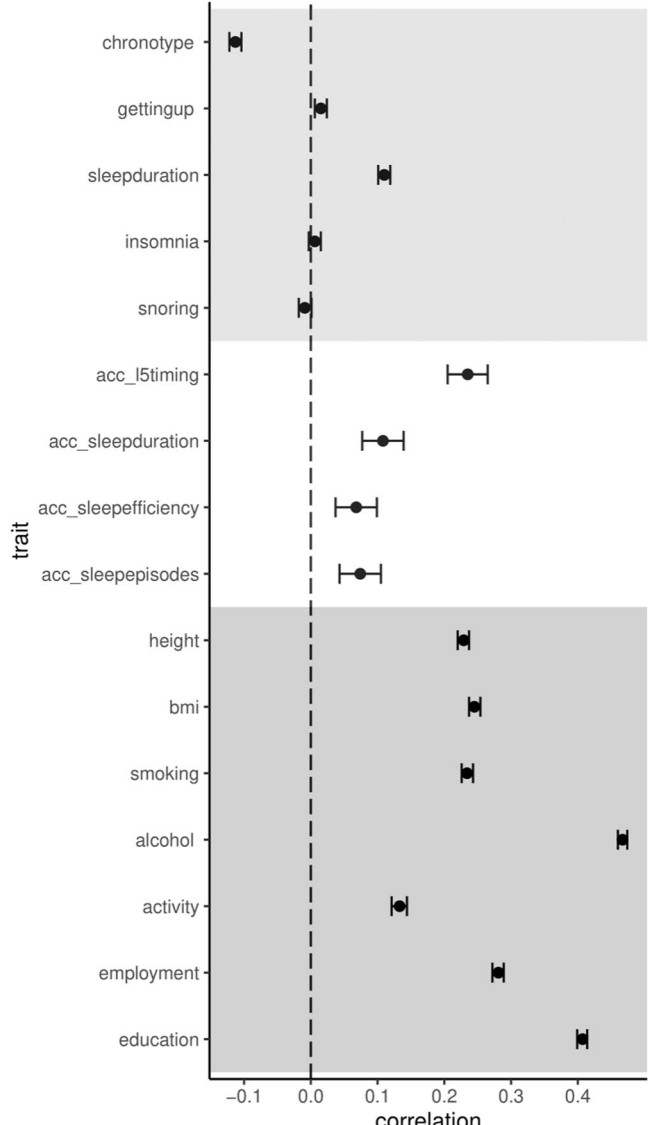

**Fig. 2 Comparison of phenotypic correlations between spouses in UK Biobank.** Acc_ = accelerometer-derived measure. Correlations and 95% confidence intervals are shown. Sample sizes are given in the accompanying source data (Supplementary Data 13).

between spouses, with the largest positive correlation between insomnia and snoring ($r = 0.07$; 0.05, 0.09), and the largest inverse correlation between sleep duration and snoring ($r = -0.06$; $-0.10$, $-0.02$) (Supplementary Fig. 3).

**Mendelian randomisation analysis.** In the UK Biobank, genetic risk scores (GRS) for each of the self-reported and accelerometer-derived sleep traits were generated based on single nucleotide polymorphisms (SNPs) surpassing genome-wide significance ($p < 5 \times 10^{-8}$) in previous genome-wide association studies (GWAS) (see "Methods"). Details of the number of SNPs contributing to the GRS and the variation explained in the sleep traits by the GRS in female and male spouses are shown in Table 3. The GRS explained between 0.1 and 1.4% of the variance in the respective sleep traits, conferring adequate genetic instrument strength for the self-reported sleep traits ($F$-statistics 132–604), although the variance explained differed between males and female spouses for the accelerometer-derived sleep traits and

**Table 2 Multivariable regression and Mendelian randomisation analysis to assess associations between sleep traits among UK Biobank spouse-pairs.**

| Sleep trait | N (pairs) | Multivariable regression | | | | | Mendelian randomisation (2SLS) | | | | | |
|---|---|---|---|---|---|---|---|---|---|---|---|---|
| | | Risk/mean difference | SE | CIL | CIU | p-value | Risk/mean difference | SE | CIL | CIU | p-value | z-test for difference |
| **Self-reported** | | | | | | | | | | | | |
| Chronotype | 47,235 | -0.113 | 0.005 | -0.122 | -0.104 | 2.42E-135 | -0.152 | 0.055 | -0.26 | -0.042 | 0.006 | 0.480 |
| Ease of waking | 47,325 | 0.015 | 0.005 | 0.005 | 0.023 | 2.00E-03 | -0.039 | 0.079 | -0.193 | 0.116 | 0.625 | 0.503 |
| Sleep duration | 47,050 | 0.111 | 0.005 | 0.102 | 0.120 | 7.97E-126 | 0.131 | 0.048 | 0.037 | 0.226 | 0.007 | 0.679 |
| Insomnia | 47,369 | 0.005 | 0.005 | -0.004 | 0.014 | 0.304 | -0.046 | 0.054 | -0.151 | 0.058 | 0.386 | 0.347 |
| Snoring | 45,546 | -0.008 | 0.004 | -0.017 | 0.000 | 0.051 | -0.154 | 0.061 | -0.274 | -0.035 | 0.011 | 0.017 |
| **Accelerometer-derived** | | | | | | | | | | | | |
| L5-timing | 3454 | 0.241 | 0.016 | 0.209 | 0.272 | 1.56E-50 | 0.486 | 0.232 | 0.032 | 0.939 | 0.036 | 0.292 |
| Sleep duration | 3454 | 0.108 | 0.016 | 0.077 | 0.140 | 1.38E-11 | 0.132 | 0.131 | -0.124 | 0.387 | 0.312 | 0.856 |
| Sleep episodes | 3454 | 0.069 | 0.016 | 0.038 | 0.100 | 1.63E-05 | -0.267 | 0.247 | -0.751 | 0.217 | 0.28 | 0.175 |
| Sleep efficiency | 3454 | 0.075 | 0.016 | 0.043 | 0.106 | 3.84E-06 | 0.024 | 0.138 | -0.246 | 0.295 | 0.859 | 0.714 |

Estimates represent the mean difference in the spouse's sleep trait (in SD) per SD increase in an individual's own sleep trait, with the exception of snoring for which estimates represent risk difference. Multivariable regression was adjusted for age at the assessment and assessment centre for both spouses. Mendelian randomisation was adjusted for age at assessment, assessment centre, genotyping chip and 10 genetic principal components (PCs) for both spouses. 2SLS = two-stage least squares.

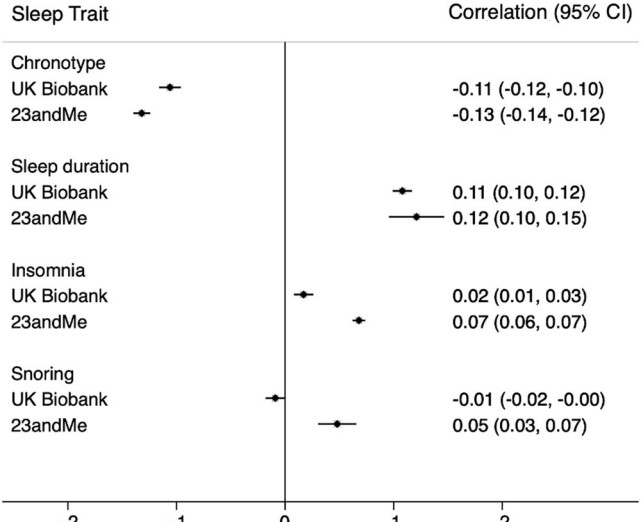

**Fig. 3 Spousal phenotypic correlations between sleep traits in UK Biobank and 23andMe.** Correlations and 95% confidence intervals are shown. Sample sizes are given in the accompanying source data (Supplementary Data 14).

there was an indication of weak instruments ($F < 10$) for sleep efficiency and L5-timing in females ($F$-statistics 4.8 and 8.5, respectively).

MR effect estimates from two-stage least squares (2SLS) analysis were largely consistent with those from MV regression (Table 2), with the exception of snoring, where MR estimated a larger inverse effect between spouses (risk difference = −0.15; −0.27, −0.04; p-value for difference from MV estimate = 0.017). We observed that participants' chronotype was more likely to induce the opposite chronotype in their spouse (mean difference = −0.15; −0.26, −0.04 SD per SD). The longer sleep duration of one spouse was positively related to sleep duration in the other (0.13; 0.04, 0.23 SD per SD). From the accelerometer assessment, activity timing was positively related between spouses in MR analysis (mean difference: 0.49; 0.03, 0.94 SD per SD) and there were consistent estimates for the effect of sleep duration, albeit with wider confidence intervals than the self-reported equivalent (0.13; −0.12, 0.39 SD per SD).

When the impact of male spouses' sleep was separated from female spouses' sleep, effect estimates were similar for the majority of sleep traits, with a few exceptions. Males' chronotype had a stronger inverse effect on the chronotype of female spouses, while females' ease of waking had a stronger inverse effect on the ease of waking of male spouses (Supplementary Fig. 4).

In addition to investigating causal estimates for one given sleep trait between spouses, we also examined cross-trait effects using MR (Supplementary Fig. 5). The directions of association between the spousal sleep traits were relatively consistent with those observed from the spousal phenotypic correlations (Supplementary Fig. 2). The strongest cross-traits effects were seen for the diurnal preference traits (ease of waking and chronotype), where easier waking was inversely related to spouses' report of morning preference (mean difference = −0.15; 0.25, −0.06 SD per SD) and reciprocally, morning preference was inversely related to spouses' ease of waking (mean difference = −0.08; 0.14, −0.01 SD per SD). However, the opposite direction of effect was observed for diurnal preference on spouses' diurnal activity, where later L5-timing was inversely related to spouses' ease of waking (mean difference = −0.30; 0.60, 0.00 SD per SD). Insomnia was also found to have a causal positive effect on spouses' reported snoring (mean difference = 0.10; 0.04, 0.16 SD per SD), but not vice versa

**Table 3 Genetic risk score (GRS) associations with sleep traits in UK Biobank.**

| Trait | N (pairs) | N SNPs | Female spouses | | Male spouses | |
|---|---|---|---|---|---|---|
| Self-report | | | Partial $R^2$ | F-statistic | Partial $R^2$ | F-statistic |
| Chronotype | 47,235 | 156 | 0.013 | 604 | 0.012 | 548 |
| Ease of waking up | 47,325 | 79 | 0.006 | 273 | 0.006 | 278 |
| Sleep duration | 47,050 | 70 | 0.006 | 279 | 0.005 | 239 |
| Insomnia symptoms | 47,369 | 40 | 0.004 | 192 | 0.003 | 161 |
| Snoring | 45,546 | 38 | 0.003 | 142 | 0.003 | 132 |
| **Accelerometer-derived** | | | | | | |
| Least-active 5 h timing | 3454 | 6 | 0.003 | 8.5 | 0.006 | 19.2 |
| Sleep duration | 3454 | 11 | 0.005 | 17.2 | 0.014 | 48.3 |
| Number of sleep episodes | 3454 | 22 | 0.014 | 47.2 | 0.010 | 33.2 |
| Sleep efficiency | 3454 | 5 | 0.001 | 4.8 | 0.005 | 17.3 |

Estimates for variance explained are adjusted for age, assessment centre, genotyping chip and top 10 principal components

(risk difference = −0.02; −0.05, 0.01). This implies that an individual with insomnia is more likely to report their spouses' snoring.

**Genetic risk score correlation**. There was limited evidence for genotypic correlations between the sleep traits as determined based on correlations of the GRS (based on ($p < 5 \times 10^{-8}$) between spouses) (−0.007 ≤ r ≤ 0.010). (Supplementary Fig. 6). Cross-trait correlations between the GRS were also less evident (−0.009 ≤ r ≤ 0.006). Findings were similar when using a series of additional GRS derived from SNPs selected at a lower $p$-value threshold from the GWAS for each sleep trait ($p < 5 \times 10^{-7}$, $p < 5 \times 10^{-6}$ and $p < 5 \times 10^{-5}$) (Supplementary Fig. 7). The only trait demonstrating consistent (but weak) evidence of correlation between the spouses was for insomnia (0.009 ≤ r ≤ 0.014).

**Effect modification**. For those sleep traits where we found evidence suggestive of effects between spouses, we investigated whether the effects were modified by a range of socioeconomic and lifestyle factors. Effect modification by age was assessed as a proxy for relationship length, whereby evidence of stronger effects with longer relationships could provide more evidence for convergence in behaviours after partnership. We assessed whether effects varied by birth location of the spouses to evaluate potential confounding by population structure (i.e., where spouses originating from similar areas may be more similar to each other than those born further apart). We also investigated whether effects were modified by employment status, children living in the household, Townsend deprivation index, household type and rural/urban location. Given the variation in the dates when the accelerometer was worn by UK Biobank participants, we also assessed whether spousal effects for the accelerometer-derived traits varied by differences in season and date of wear between the spouses. In addition, we look at whether spouses were more likely to sleep concordantly if they had similar activity patterns during wake time using a measure of M10-timing, which is accelerometer-derived timing of the most active 10 h of the day.

There was limited evidence for modification of the chronotype effect by any of the factors considered ($I^2 = 0\%$, $P_{Het} \geq 0.45$). Moderate heterogeneity by mean age was observed for sleep duration ($I^2 = 62\%$, $P_{Het} = 0.07$) with larger effects at older ages (mean differences = −0.02; −0.18, 0.13 SD per SD for 40–54 years; 0.17; 0.05, 0.39 SD per SD for 55–61 years; 0.22; 0.07, 0.36 SD per SD for 62-70 years). There was moderate heterogeneity by age difference for snoring ($I^2 = 61\%$, $P_{Het} = 0.07$) and by Townsend deprivation index ($I^2 = 57\%$, $P_{Het} = 0.10$) but no linear trend was observed. There was also evidence to suggest that activity timing effects (based on accelerometer-derived L5-timing) were stronger

in spouses when there were no children in the household (mean difference = 0.90; 0.20, 1.60 SD per SD with no children vs. −0.11; −0.82, 0.62 SD per SD with one or more child in the household; I2 = 74%; $P_{Het} = 0.05$) (Supplementary Fig. 8).

**Robustness of MR analyses: horizontal pleiotropy**. For those sleep traits where there was an indicated effect between spouses, we found little evidence for horizontal pleiotropy based on: (i) a Sargan test which evaluates between-SNP heterogeneity in the causal estimates (Supplementary Data 6) and (ii) an MR-Egger intercept test which tests for directional pleiotropy (Supplementary Data 6). We also evaluated effect estimates using methods which can account for pleiotropy in this setting[37]. Effect estimates were largely consistent in direction with those obtained from both MV and 2SLS analysis, although with wider confidence intervals for the MR-Egger and Least Absolute Deviation (LAD) approaches, which crossed the null (Fig. 4 and Supplementary Data 7). However, mean F-statistics and $I^2$ values for the individual SNP-exposure estimates used in these analyses were found to be small, indicating the presence of weak genetic instruments (Supplementary Data 6). Together, findings suggest that our main results are likely robust to horizontal pleiotropy, although the presence of weak instruments indicates that this sensitivity analysis should be interpreted with caution.

**Robustness of MR analyses: Winner's curse**. We derived GRS comprising a subset of SNPs used in the main analysis which replicated in independent datasets in order to evaluate potential Winner's curse. This could be present due to an overlap between the sleep GWAS and spouse-pair sample, leading to an over-estimation of the individual SNP effects on the exposure. Effect estimates for chronotype and sleep duration using replicated SNPs were largely consistent with those from the main analysis (Supplementary Data 8). For insomnia, the effect estimates obtained based on SNPs which replicated in 23andMe were more consistent with a positive effect between spouses (0.173; −0.025, 0.371), with estimates in the opposite direction to those obtained in the main analysis from the UK Biobank (−0.046, −0.151, 0.058) (Supplementary Data 8). We also re-estimated the effects of insomnia using SNPs identified in a meta-analysis of UK Biobank and 23andMe[38]. Estimates were also in the opposite direction to the main analysis although still consistent with the null (0.076, −0.025, 0.177) (Supplementary Data 9)

**Discussion**
Using a large sample of spouse pairs within the UK Biobank study, we investigated the correlation between sleep patterns and circadian preference between spouses with data on both self-

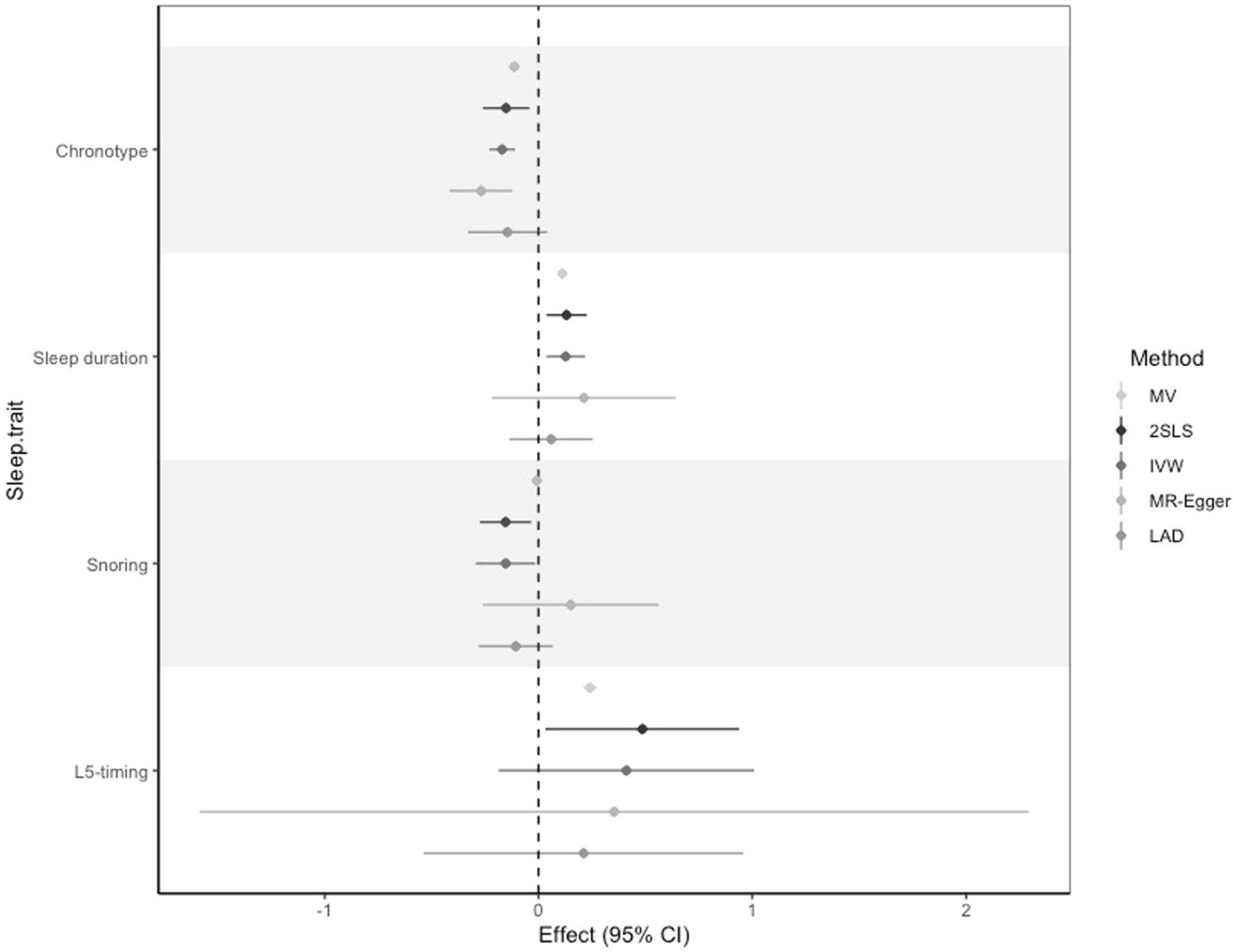

**Fig. 4 Comparison of multivariable and Mendelian randomisation estimates in UK Biobank.** MV multivariable regression, 2SLS two-stage least squares, IVW inverse-variance weighted, MR-Egger MR-Egger approach, LAD least absolute deviation. Estimates represent the mean difference in the spouse's sleep trait (in SD) per SD increase in an individual's own sleep trait, with the exception of snoring for which estimates represent risk difference. 95% confidence intervals are shown. Sample sizes are given in the accompanying source data (Supplementary Data 15).

reported and accelerometer-derived sleep traits. We found evidence for weak positive phenotypic correlations between spouses for sleep duration, ease of waking, timing of diurnal activity and number of nocturnal sleep episodes as well as a weak inverse correlation between spouses for reported chronotype (diurnal preference). Findings of a positive correlation for sleep duration and inverse correlation for chronotype were replicated in 23andMe. Several cross-trait correlations were also observed between spouses. Individuals who reported having insomnia were more likely to report snoring by their spouse in both UK Biobank and 23andMe. Additionally, morning preference in the index individual was associated with later activity timing of their spouse in UK Biobank. In MR analyses, positive effects of sleep duration and activity timing were found between spouses as well as inverse effects of chronotype and snoring on the same traits in their partners. We were unable to determine whether males or females had more bearing on their spouse's sleep patterns, and no large differences in effects were found by socioeconomic, demographic and lifestyle factors or accelerometer characteristics. This was except for the accelerometer-derived activity timing, where the later diurnal activity of one spouse had a larger effect on the other if spouses lived in households without children. In addition, for sleep duration, larger positive effects were observed at older ages,

suggesting a convergence in sleep duration between spouses over time. GRS correlations in the sleep traits were weaker than phenotypic correlations between spouses, providing some evidence against assortative mating (whereby individuals select phenotypically similar partners).

In line with our findings, Randler and Kretz found correlations in several sleep-wake variables between spouses[39]. However, in contrast to the moderate positive relationship in chronotype observed in that small study ($r = 0.40$, $n = 84$ couples), weak inverse effects for chronotype were found between spouses in the UK Biobank and 23andMe ($r = –0.11$ and $–0.13$, respectively), which was supported by MR analysis. The inverse correlation is unexpected, especially given the positive correlation between accelerometer-assessed L5-timing. This suggests a separation of subjective chronotype, reflective of diurnal preference, from actual objective sleep timing. Inverse correlations also go against the plethora of evidence indicating widespread similarities (rather than differences) of spouses for many phenotypes (Fig. 2)[33]. One possibility is that in the subjective appraisal of chronotype a natural referent is one's spouse, whereby any difference between spouses (e.g., if one tends to get up slightly earlier than the other) may be amplified (e.g., more likely to report being a 'morning person' than the other) and therefore induce an inverse

correlation. In contrast, traits such as sleep duration, even if self-reported, are perhaps less likely to be influenced by this dyadic comparison. However, it should be emphasised that the inverse correlation observed for chronotype was relatively weak in both studies.

Similar to our findings, the previous study did not find a correlation between the length of the relationship (proxied in our study by mean age of couples) and dissimilarity in morning-evening preference[39]. They interpreted this as suggesting initial assortment by chronotype, whereas the lack of GRS correlation ($r = 0.001$) and discordance in chronotype between spouses does not provide the same evidence for assortative mating in UK Biobank and 23andMe. Additionally, our findings suggest that activity timing, as proxied by accelerometer-assessed L5-timing in UK Biobank, converges between spouses and so may deviate from reported diurnal preference after partnership.

The literature regarding insomnia between spouses is less consistent, with some studies reporting a protective effect of being in a partnership on insomnia risk[3] and others showing more frequent wake transmissions among partners of individuals with insomnia[40]. Another study found that while actigraphy-assessed sleep movements were greater when couples were sleeping together, subjective sleep was generally reported to be worse when sleeping apart[20]. We did not find a strong correlation of insomnia symptoms between spouses, assessed based on self-report as well as accelerometer measures of sleep efficiency and number of nocturnal sleep episodes. However, we did find some evidence for an effect of sleep duration between spouses.

We also observed an inverse relationship between spouses' reported snoring and a positive relationship between snoring and insomnia in the spouse. Spouses of snorers have been found to more frequently report sleeping problems including insomnia[23]. However, the results of the MR analysis suggested that this association may reflect a positive effect of an index individual's reported insomnia on snoring in their spouse, rather than the spouse's snoring inducing insomnia. This may be explained by the fact that reported snoring is captured via the spouse of the snorer in the UK Biobank question, "Does your partner or a close relative or friend complain about your snoring", and so if an individual experiences insomnia symptoms, they may be more likely to notice and report snoring in their spouse.

Most of the studies which have investigated sleep correlations between spouses have done so in small, cross-sectional settings, typically with fewer than a hundred couples[20,21,39,40]. The present study uses data on 47,420 determined spouse-pairs within the UK Biobank to evaluate the correlation in sleep traits between spouses, with replication in 23andMe ($n \leq 127,035$ spouse-pairs), as well as the use of genetic analysis to evaluate causal effects underlying spousal correlation in sleep traits. Furthermore, the availability of accelerometer data on ~3500 couples in UK Biobank has enabled a comparison of both subjective reported sleep traits and objective sleep measures between spouses. Unlike findings from a previous study which used a similar genetic approach in the UK Biobank to infer assortative mating on both height and alcohol consumption[34], our finding of low GRS correlation between sleep traits suggests that correlation for sleep traits did not exist prior to cohabitation but that spouses may influence each other's sleep patterns after partnership.

The current study suffers from some limitations with respect to both the phenotypic measures and genetic analysis which require discussion. We were unable to directly obtain information on spouses within UK Biobank and instead, spouses were inferred based on several criteria, including marital status and location. Other studies have used similar methods to determine spouses and the validity of the derived spouse-pair sample has been previously verified[34]. In 23andMe, a different method was used to infer spouse pairs, based on genetic trios to obtain mother-father pairs. However, we could not determine whether the 23andMe 'parents' were separated and not living together. We also did not have information on whether couples in the UK Biobank or 23andMe shared a bed, so we are unable to determine the extent to which the effects observed are directly attributed to bed sharing rather than cohabitation.

Self-reported measures of the sleep traits in UK Biobank and 23andMe were based on a limited number of questions which may crudely assess underlying sleep traits and/or be subject to bias. However, there is a trade-off between the number of individuals with sleep phenotypes obtained from easy-to-administer questionnaires and more objective or clinically derived sleep measures, which are typically available to a smaller number of participants. The replication of genetic associations identified in relation to the self-reported sleep traits in UK Biobank with comparable phenotypes in independent studies including 23andMe[38,41,42], and with objectives measures, i.e., from the accelerometer assessment in UK Biobank[36,42], serves as an indirect means of validation of the self-report measures[43]. Further to this, we also directly assessed spousal concordance using accelerometer-derived measures in a subset of the UK Biobank.

Although the availability of accelerometer measures enabled an objective assessment of sleep patterns between a large sample of spouses in the UK Biobank, individuals did not wear an accelerometer at the same time as the self-reported assessment, with a median time difference of 6 years between assessments for the spouses. Additionally, spouse pairs wore the accelerometer 7 months apart on average, with only 4% of individuals wearing an accelerometer at the same time as their spouse.

While the results of our analysis suggest potential interactions after partnership which may result in convergence of sleep traits over time, we were unable to investigate this longitudinally, which would require repeat assessments of sleep traits. While we investigated effect modification by age, this is a crude proxy measure for relationship length. Furthermore, while the small PRS correlations between spouses were used to provide evidence against strong assortative mating based on the sleep traits, weak assortative mating effects may still exist which are cumulatively evident at the phenotypic level but were not captured by the subset of variants included in this analysis.

The use of MR allowed us to overcome problems of confounding and reverse causation and has enabled an assessment of the potentially causal relationship in sleep traits between spouses. While this offers additional inference to phenotypic correlations, several other assumptions must be made in order for the causal estimates to be valid[24–26]. We have attempted to address and overcome most of these assumptions, with assessments of instrument strength, population stratification, horizontal pleiotropy and Winner's curse. Most robust inferences can be made when the estimated effects are consistent in sensitivity analyses which attempt to address these assumptions, which was the case for effects observed in relation to chronotype, diurnal behaviour and sleep duration. While the genetic instruments were found to be strongly related to the sleep traits, for some of the accelerometer-derived traits, $F$-statistics were small which could indicate weak instrument bias. This was particularly the case for accelerometer traits in females, suggesting that there may be some sex differences in the genetic contribution to the sleep traits. While we used GRS derived from SNPs identified in GWAS of men and women combined, GRS comprising SNPs could be generated from sex-specific GWAS which may serve as stronger instruments. However, there would be a necessary trade-off with lower sample sizes for the sex-specific rather than sex-combined GWAS, which may reduce statistical power. Furthermore, mean $F$-statistics of the individual SNP effects used in the analyses

accounting for pleiotropy were found to be small. However, the effects using the inverse-variance weighted approach were very consistent with those obtained using the stronger GRS instrument in 2SLS analysis, indicating that this bias is unlikely to be a major contributing factor.

We found that the spouse-pairs in UK Biobank differed in several sociodemographic characteristics from the remainder of the UK Biobank cohort (including those participants who reported living with a spouse but whose spouse was not in the study). While these differences were marginal, they suggest that the spouse-pairs, particularly those with accelerometer data, were healthier and more affluent which may influence the generalisability of findings to the full cohort. The UK Biobank and 23andMe spouse-pairs are also unlikely to be fully representative of the general population, being more highly educated, more affluent and in better health on average[44]. Additionally, the mean ages of the spouses in both studies were 57 and 63 years old, and so the findings regarding the correlation in sleeping patterns and circadian preferences between spouses may not necessarily extrapolate to younger couples, particularly those with contrasting work schedules. Furthermore, factors influencing selection into the UK Biobank and 23andMe[45], and spousal assortment on other traits[19], may bias spousal comparisons in both observational and MR analysis. However, previous simulations have suggested that this most likely results in a bias towards the null, leading to an underestimation of the true effect.

Another selection factor which could have biased estimates in the present study is relationship dissolution, whereby sleep concordance/discordance could influence the likelihood of remaining in a relationship and of participating in the UK Biobank study together. The lack of evidence to suggest that the mean age of each spouse-pair (as a proxy for relationship length) was associated with sleep correlation suggests that the effects observed are unlikely to be due to relationship dissolution. However, further work is required to investigate whether similarities/dissimilarities in spousal sleep traits are predictive of relationship dissolution[46].

Within two large population-based resources comprising a high proportion of spouse pairs, we established correlations between several sleep traits between spouses. Within the UK Biobank, we were also able to evaluate accelerometer-based sleep assessment and used a genetic analysis to demonstrate the effects of an individual's sleep trait on that of their spouse for chronotype, diurnal activity, sleep duration and snoring. Weak cross-trait associations were also evident in the study. Our results suggest that these effects may be due to interaction after partnership rather than confounding by social homogamy or assortative mating. According to the US National Sleep Foundation, 76% of adults with a sleep disorder share a household with at least one other person who does[47]. Our findings provide insights into sleep behaviour among co-habiting spouses and highlight how certain sleep traits can be influenced by the sleep of a person's spouse. This helps us gain a better understanding of the aetiology of poor sleep, which may in turn impact relationship factors and could have further downstream physical and mental health consequences[6,7,9,10]. The results of this study promote further investigation into the familial impacts of sleep and sleep-related ill health and raise possible opportunities for sleep interventions aimed at the family level. However, the magnitude of sleep effects was small and whether this level of correlation between spouses contributes towards disease risk, as indicated in ref. [48], requires further investigation.

## Methods
**UK Biobank: study description.** The UK Biobank is a population-based cohort study consisting of >500,000 participants, aged between 40 and 70 years, who were recruited between 2006 and 2010 in the UK[49]. At recruitment, the participants gave informed consent to participate and be followed up. UK Biobank has received ethical approval from the UK National Health Service's National Research Ethics Service (ref 11/NW/0382).

**UK Biobank: genetic data.** The full data release in the UK Biobank contains the cohort of successfully genotyped individuals ($N = 488,377$). A total of 49,979 individuals were genotyped using the UK BiLEVE array and 438,398 using the UK Biobank axiom array. Pre-imputation quality control, phasing and imputation of the UK Biobank genetic data have been described elsewhere[50]. We restricted the dataset to a subset of 463,827 individuals of recent European descent with available genotype data, with individuals of non-European descent removed based on a k-means cluster analysis on the first four genetic PCs[51].

**UK Biobank: household composition.** At baseline assessment, participants were asked to report the number of people living in their household (including themselves). For those who reported more than one person, they were asked how the other people were related to themselves, or whether they were unrelated. Using this information, we determined three groups of participants: living with a spouse, living with someone other than a spouse, and living alone.

**UK Biobank: spouse-pair sub-sample.** Using the European sub-sample, spouse-pair information was determined using the same approach described previously[34]. In brief, household sharing information was used to extract pairs of individuals who (a) report living with their spouse, (b) report the same length of time living in the house, (c) report the same number of occupants in the household, (d) report the same number of vehicles, (e) report the same accommodation type and rental status, (f) have identical home co-ordinates (to the nearest 1 km) and (g) are registered to the same UK Biobank recruitment centre and h) both have available genotype data. Exclusions were made if more than two individuals shared identical information across all variables (and so spouses could not be clearly defined), if potential couples who were the same sex (as our analysis was related to sex differences in sleep patterns and hence effects in heterosexual couples), if couples reported the same age of death for both parents (suggesting they were siblings rather than spouses), and if estimated genetic relatedness was deemed to be too high (identify-by-descent (IBD) > 0.1, suggesting siblings or parent-child pairs rather than spouses). The final sample included 47,549 spouse pairs.

**UK Biobank: sleep questionnaire measures.** At baseline assessment, participants were given a touchscreen questionnaire, which included questions about sociodemographic status, lifestyle and environment, early life and family history, health and medical history, and psychosocial factors. This included several questions related to sleep and circadian traits (Supplementary Note 1).

We assessed spousal correlations between 5 self-reported sleep traits: chronotype (morning/evening preference), ease of waking up, insomnia symptoms, sleep duration and snoring. Chronotype was coded into five categories ("Definitely an 'evening' person", "More an 'evening' than a 'morning' person", "Do not know", "More a 'morning' than 'evening' person", "Definitely a 'morning' person"); ease of waking was coded into four categories ("Not at all easy", "Not very easy", "Fairly easy", "Very easy"); total 24-h sleep duration was reported in whole hours; insomnia symptoms frequency was coded into three categories ("Never/rarely", "Sometimes", "Usually") and snoring was coded as a binary

variable ("No" or "Yes"). Those who responded "Do not know" or "Prefer not to say" were treated as missing data for all sleep traits, except for chronotype where "Do not know" was treated as an intermediate category. A binary was also generated for chronotype ("Evening vs. Morning person"), by combining definite and intermediate categories, and excluding those who reported "Do not know", as well as for insomnia, by combining "Never/rarely" and "Sometimes"). This was done to aid comparison with findings from 23andMe.

**UK Biobank: accelerometer measures**. A triaxial accelerometer device (Axivity AX3 was worn between 2.8 and 8.7 years after the study baseline by 103,711 individuals from the UK Biobank for a continuous period of up to 7 days. Details of data collection and processing have been previously described[52]. Measures of sleep quality, quantity and timing have been derived by processing raw accelerometer data with the use of the open-source R package GGIR[53]. More details on the accelerometer-based sleep measures derived in the UK Biobank can be found in ref. [36]. These have been returned to UK Biobank as part of data return 1862.

We investigated four continuous measures: mean L5 time (midpoint of least-active 5 h), mean number of nocturnal sleep episodes, mean daily sleep duration and mean sleep efficiency. The least-active 5 h (L5) of each day was derived using a 5-h period of minimum activity. This period was estimated using a rolling average of the respective time window and defined as the number of hours elapsed from the previous midnight. The sleep period time (SPT)-window was estimated using an algorithm described in ref. [54]. The number of sleep episodes within the SPT window was defined as the number of sleep bouts separated by at least 5 min of wakefulness within the SPT window. The summed duration of all sleep episodes was used as an indicator of sleep duration within the SPT window. Sleep efficiency was calculated as sleep duration divided by SPT-window duration.

We excluded individuals flagged by UK Biobank as having data problems (field 90002), poor wear time (field 90015), poor calibration (field 90016), or unable to calibrate activity data on the device worn itself requiring the use of other data (field 90017). Individuals were also excluded if the number of data recording errors (field 90182), interrupted recording periods (field 90180), or duration of interrupted recoding periods (field 90181) was greater than the respective variable's 3rd quartile + 1.5 × IQR, as previously reported[36].

**UK Biobank: covariates**. The age of the participants at the baseline assessment (when self-reported measures were obtained) was derived based on their date of birth and the date of attending the assessment centre. Age at the accelerometry assessment was estimated using the date of birth and the date of the first recording day. Sex was determined at recruitment and individuals with sex mismatch (derived by comparing genetic sex and reported sex) ($n = 378$) or individuals with sex-chromosome aneuploidy ($n = 652$) were excluded from the analysis. Information on which of the 22 centres in Scotland, England and Wales where assessments were undertaken was also obtained. Place of birth in the UK was ascertained from a verbal interview at the assessment centre and UK Grid co-ordinates (north and east) were determined. The season when the accelerometer was worn was also ascertained. For all genetic analysis, we also included a genotyping chip and the top 10 PCs derived from the genetic data as covariates.

**23andMe: study description**. Individuals in the 23andMe dataset were customers of 23andMe, Inc., a personal genomics company. Participants provided informed consent and participated in the

research online, under a protocol approved by the external AAHRPP-accredited IRB, Ethical & Independent Review Services (E&I Review). Participants were included in the analysis on the basis of consent status as checked at the time data analyses were initiated.

**23andMe: spouse-pair sub-sample**. Parent-offspring trios were identified from the 23andMe database using identity-by-descent (IBD) information. Specifically, a segment-based approach was used to designate IBD1 and IBD2, corresponding to regions that have the indicated number (one or two) of shared haplotypes between two individuals. Every individual was considered as a potential child in a trio (called index individual), and candidate parents were identified as those sharing at least 42.5% of their genome IBD1 and no more than 10% of their genome IBD2 with the index individual. Genome-wide relatedness between pairs of candidate parents was then computed to eliminate incorrect candidate pairs (for example, where one candidate parent is a true parent of the index individual and the other candidate parent is a child of the index individual) by requiring candidate parents to share no more than 20% of their genome IBD1 with each other. Lastly, trios were checked for Mendelian concordance on 100 random SNPs with a genotyping rate of 99.9% and MAF > 0.3 and were required to be concordant on at least 95 of the 100 tested SNPs.

From 771,487 parent-offspring trios in the 23andMe research cohort, children were removed to obtain mother-father pairs. The trios were subsetted to 531,856 pairs, where both parents had complete data for the sleep trait, age and sex, and were of predominantly European ancestry. A detailed description of the 23andMe ancestry classifier can be found here[55,56] where participants defined as predominantly European ancestry were those who, after 23andMe ancestry composition, had a probability of European + Middle Eastern ancestry > 0.97% or European ancestry > 0.90%.

**23andMe: sleep questionnaire measures**. As part of the personal genomics service, all customers are invited to participate in research, which occurs predominantly through web-based research surveys. Participants are asked a number of questions about their sleep habits.

We assessed spousal correlations between 4 self-reported sleep traits: chronotype (morning/evening preference), insomnia symptoms, sleep duration and snoring. Research participants were asked, "Are you naturally a night person or a morning person?", with options "Night person", "Morning person", "Neither", "It depends" and "I'm not sure". A binary variable was generated as "Morning person" vs. "Night person", where "It depends" and "I'm not sure" were treated as missing data. Participants also answered the question, "Have you ever been diagnosed with, or treated for, insomnia?" with options "Yes", "No", and "I'm not sure". A binary variable was generated as "Yes" vs. "No", where "I'm not sure" was treated as missing data. Participants were next asked, "During the past month, how many hours of actual sleep did you get at night? (This may be different than the number of hours you spend in bed.)" Responses were integers, with extreme responses of less than 3 h or more than 12 h excluded. Finally, participants responded to the question, "On most nights, do you snore", with options "Yes", "No" and "I'm not sure". A binary variable was generated as "Yes" vs. "No", where "I'm not sure" was treated as missing data.

**23andMe: covariates**. Accompanying surveys provide self-reported data on covariates such as age and sex. Ancestry composition was performed as previously reported[56]. Inclusion was

restricted to individuals of predominantly European ancestry to minimise confounding by ancestry.

**Statistics and reproducibility**. Phenotypic correlations in sleep traits between spouses were calculated in both UK Biobank ($n = 47,420$ spouse-pairs with sleep measures) and 23andMe ($n = 127,035$ spouse-pairs with sleep measures), for which estimates were compared. The Mendelian randomisation analysis involved participants from UK Biobank only ($n = 47,420$ spouse-pairs with sleep measures), with supplementary analysis (genetic risk score correlation, effect modification and an assessment of MR assumptions and bias) performed to evaluate the nature and robustness of the effects observed.

**Phenotypic correlations**. To evaluate the phenotypic correlation of sleep traits in the UK Biobank, we compared self-reported sleep traits and accelerometer measures between spouses. We estimated the spousal correlation for the 5 self-reported sleep traits and 4 accelerometer-based sleep traits by assessing the correlation between the relevant variable for an individual against the relevant variable for their spouse, using a Pearson correlation test adjusting for age, spouse's age and recruitment centre. With one unique phenotype pairing within couples (male sleep trait/ female sleep trait), each individual in the dataset was included only once as either the reference individual or their spouse. To contextualise the findings, we also calculated the correlation between spouses for height (field 12144), body mass index (field 21001), smoking status (field 20116), alcohol intake (field 1558), physical activity (field 894), employment status (field 6142) and education level (field 6138), obtained at baseline assessment.

To assess the spousal correlation for the 4 self-reported sleep traits in 23andMe, a regression of the sleep trait on age was first performed and then the residuals were extracted for each parent. A Pearson correlation test for the age-corrected values was used to assess the correlation between the relevant variable for an individual against the relevant variable for their spouse. With one unique phenotype pairing within couples (male sleep trait/female sleep trait), each individual in the dataset was included only once as either the reference individual or their spouse.

**Mendelian randomisation analysis**. We used MR to investigate the evidence for an effect of an individuals' sleeping patterns on the sleeping patterns of their spouse in UK Biobank. This was done by generating a series of genetic risk scores (GRS) for the 5 self-reported sleep traits (chronotype (morning/evening preference)[41], ease of waking up, insomnia symptoms[57], sleep duration[42] and snoring[58]) and 4 accelerometer-derived sleep traits (mean L5 time, number of nocturnal sleep episodes, mean daily sleep duration and mean sleep efficiency[36]). Scores were generated based on single nucleotide polymorphisms (SNPs) surpassing genome-wide significance ($p$-value $< 5 \times 10^{-8}$) in relation to the sleep traits in genome-wide association studies (GWAS) ($n = 85,670 – 461,569$). SNP lists were obtained from the relevant GWAS summary statistics available via the Sleep Disorder Knowledge Portal[59]. These were pruned for linkage disequilibrium ($r^2 = 0.001$) based on a European reference panel, using the clump_data function from the "TwoSampleMR" package in R (version 3.5.1)[60]. Detailed information on the genetic variants is given in Supplementary Data 10.

The genetic variants were extracted from the UK Biobank genetic data and unweighted GRS were generated as the total number of sleep trait–increasing alleles (morning preference alleles for chronotype) present in the genotype of each participant. Two-stage least squares (2SLS) instrumental variable

analyses were performed between standardised sleep traits (all traits with mean 0 and SD 1) with adjustment for age at assessment, assessment centre, genotyping chip and 10 genetic principal components (PCs) to minimise confounding by population stratification. This was performed using each GRS as an instrument for its respective sleep traits using the "ivreg2" command in Stata (version 15).

Standardised variables are presented in the table to allow for direct comparisons with the correlation coefficients estimated for the phenotypic correlation. To enable this, ordinal variables were treated continuously. We also performed MV regression using the same variables with adjustment for age at the assessment and assessment centre and performed a $z$-test for difference with the 2SLS estimate to determine the extent to which the effects estimated from MR were consistent with the observational associations.

With two unique pairings between genotype and sleep trait in each couple (male spouse genotype/female spouse sleep trait and the converse), each individual in the dataset was included twice as both the reference individual and as the spouse. This analysis was performed by sex (i.e., male spouse genotype/female spouse sleep trait and female spouse genotype/male spouse sleep trait) to evaluate any differential effects between males and females on their spouses' sleep patterns and then combined the estimates obtained using an inverse-variance weighted random effects meta-analysis.

**Genetic risk-score correlation**. We assessed the correlation between the sleep GRS across spouse-pairs in UK Biobank, adjusting for age, spouse's age, assessment centre, genotyping chip and top 10 PCs. With one unique genotype pairing within couples (male spouse genotype/female spouse genotype), each individual in the dataset was included only once as either the reference individual or their spouse. In sensitivity analysis, we also assess the spousal correlation of three additional GRS for each sleep trait, derived using less stringent $p$-value thresholds for selecting contributing SNPs ($p < 5 \times 10^{-7}$, $p < 5 \times 10^{-6}$ and $p < 5 \times 10^{-5}$).

**Effect modification**. Where a sleep trait was found to have an effect on the same sleep trait of the spouse in UK Biobank, we investigated the extent to which this estimate varied by various socioeconomic, demographic and lifestyle factors and accelerometer characteristics. This included the mean age of the spouses, the difference in ages between the spouses, the birth location of the spouses, the employment status of the spouses, the presence of children in the household, the Townsend deprivation index, household type and rural/urban location. For any accelerometer measures, we also investigated differences between spouses in terms of the season of wear, time difference in wear and activity patterns during wake time.

We investigated whether spousal effects were modified these factors by repeating MR analyses in subgroups as follows: (1) by thirds of the age distribution; (2) by less or more than 100 km from where their spouse was born[34]; (3) by whether the spouses were both employed, one employed and one unemployed/retired, or both unemployed/retired; (4) by presence of absence of children in the household; (5) by thirds of the Townsend deprivation index, an area-based score of social deprivation; (6) by whether they live in a house/bungalow or flat/maisonette/ apartment, and (7) by rural or urban area classification. Both (5) and (7) were determined from the postcode of the spouses immediately prior to joining UK Biobank.

For the accelerometer measures, we also assessed effects based on the following subgroups: (1) whether the season of wearing the accelerometer differed between spouses (yes or no); (2) by thirds of the difference in accelerometer wear-time between spouses; (3) by thirds of the difference in M10-timing between spouses, which is the accelerometer-derived timing of the most active 10 h of the day. We meta-analysed estimates assuming a fixed effects model using the meta package in R (version 3.5.1) to obtain an $I^2$ value for heterogeneity.

**Assessing MR assumptions and evaluating bias**. MR analysis requires various assumptions to be satisfied in order for effects to be estimated: (1) that the genetic instrument is robustly associated with the exposure (instrument strength); (2) that the genetic instrument is independent of potential confounders of the exposure-outcome association (no confounding) and (3) that the genetic instrument influences the outcome exclusively through its effect on the exposure (no horizontal pleiotropy). Various steps were taken to assess these assumptions, as outlined below.

Partial $r^2$ values and $F$-statistics from the first-stage regression between each GRS and the index individuals' sleep traits were examined to check adequate instrument strength.

While genetic variants should not theoretically be related to potential confounding factors, concerns about potential violation of this assumption relate to confounding by ancestry or population stratification, including assortative mating effects. To address this, we adjusted for principal components derived from the genetic data in the MR analysis in order to control for population structure. The sensitivity analysis examining spousal correlation by geographic birth proximity was also used to evaluate potential confounding by social homogamy. We also examined the influence of assortative mating by evaluating the GRS correlation of the sleep traits, as described above.

Horizontal pleiotropy, where genetic variants may influence the outcome of interest through pathways other than via the exposure, is an important limitation in conventional MR analysis. However, in the context of spousal effects, pleiotropy of the genetic variants is arguably less problematic since there are unlikely to be biological mechanisms by which an individual's genotype could plausibly affect their spouses' phenotypes other than via their own phenotype. Nonetheless, there may be other (e.g., social) mechanisms which give rise to pleiotropy of the variants, and so we have conducted sensitivity analyses to evaluate this.

To assess bias due to horizontal pleiotropy, we first explored between-SNP heterogeneity using the Sargan over-identification test. We also applied a method that estimates unbalanced horizontal pleiotropy in a one-sample MR setting[37]. This method provides causal estimates using methods which have been adapted from the two-sample MR setting, including inverse-variance weighted (IVW) meta-analysis[61], MR-Egger[62] and least absolute deviation (LAD) regression (similar to the weighted median approach[63]). More details of these methods are described in ref. [37].

Winner's curse can occur when the study in which the genetic variants were identified at genome-wide significance ($p < 5 \times 10^{-8}$) is the same as the one used to perform the MR analysis. Since the genetic variants for the sleep traits were predominantly identified in the UK Biobank, this may bias causal estimates towards the null. To minimise the impact of Winner's curse, we used unweighted GRSs in the main MR analysis, rather than those weighted by the effect estimates obtained in the GWAS. We also performed MR using GRS comprising those genetic variants that replicated in independent datasets for chronotype[41], insomnia[38] and sleep duration[42]. Replication was determined based on

genome-wide significance in 23andMe for chronotype ($n = 248,100$) and insomnia ($n = 944,477$), and $p < 0.05$ in CHARGE given the smaller sample size of this replication dataset ($n = 47,180$). Information on the genetic variants used is described in Supplementary Data 11.

For insomnia, the SNPs used to assess Winner's curse were determined from a different GWAS of insomnia to that used in the main analysis[57] and comprised a meta-analysis of UK Biobank and 23andMe[38,64]. In an additional sensitivity analysis, we performed MR analysis using a GRS derived from a larger number of SNPs identified at genome-wide significance in the meta-analysis and compared estimates. Information on the genetic variants used is described in Supplementary Data 12.

**Reporting summary**. Further information on research design is available in the Nature Portfolio Reporting Summary linked to this article.

## Data availability

Summary-level data for UK Biobank and 23andMe are fully disclosed in the manuscript. SNP lists were obtained from the relevant GWAS summary statistics available via external repositories[59,64]. Individual-level data are not publicly available due to participant confidentiality and in accordance with the IRB-approved protocol under which the study was conducted. Source data underlying Figs. 2–4 are available in Supplementary Data 13–15. For details on accessing the source data from these studies, please contact access@ukbiobank.ac.uk and apply.research@23andme.com.

## Code availability

The code used to estimate phenotypic and genetic correlations, and to perform the main one-sample Mendelian randomisation analysis in UK Biobank, is available via GitHub (https://github.com/rcrichmond/spousal_sleep)[65].

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

## Acknowledgements

This research was conducted using the UK Biobank Resource under application number 16391. We thank the participants and researchers from the UK Biobank who contributed or collected data. We would also like to thank the research participants and employees of 23andMe for making this work possible. R.C.R., L.H., J.L., D.A.L. and G.D.S. are members of the MRC Integrative Epidemiology Unit at the University of Bristol funded by the Medical Research Council (MM_UU_00011/1 and MC_UU_00011/6). RCR is supported by the CRUK-Integrative Cancer Epidemiology Programme (C18281/A29019). K.H. is supported by a Humboldt Research Fellowship from the Alexander von Humboldt Stiftung. J.L. is funded by Novo Nordisk as a Novo Nordisk - Oxford Post-doctoral Research Fellow. D.A.L. is a British Heart Foundation Chair (CH/F/20/90003) and NIHR Senior Investigator (NF-0616-10102). This study was supported by the NIHR Biomedical Research Centre at the University Hospitals Bristol NHS Foundation Trust and the University of Bristol and the NIHR Manchester Biomedical Research Centre. The views expressed in this publication are those of the authors and not necessarily those of the National Health Service, National Institute for Health Research, or Department of Health and Social Care.

## Author contributions

R.C.R. conceived the study and conducted the main analysis. K.H. conducted the replication analysis in 23andMe and J.L. assisted with sensitivity analyses. R.C.R. and C.V. drafted the initial manuscript. L.J.H., K.H., S.J., J.L., the 23andMe research team, X.W., M.N.W., M.K.R., D.A.L. and G.D.S. assisted with interpretation, commented on drafts of the manuscript, and approved the final version. R.C.R. is the guarantor and attests that all listed authors meet authorship criteria and that no others meeting the criteria have been omitted.

## Competing interests

R.C.R. is an Editorial Board Member for *Communications Biology*, but was not involved in the editorial review of, nor the decision to publish this article. X.W. and the 23andMe Research Team are employees of 23andMe, Inc. and own stock and/or stock options in 23andMe, Inc. K.H. is a former employee of 23andMe, Inc. and holds stock and/or share options in 23andMe, Inc. L.H. is a current employee and stockholder of GlaxoSmithKline. C.V. is a current employee of IQVIA GmbH. The remaining authors declare no competing interests.

## Additional information

---

## 23andMe Research Team

Stella Aslibekyan[8], Adam Auton[8], Elizabeth Babalola[8], Robert K. Bell[8], Jessica Bielenberg[8], Katarzyna Bryc[8], Emily Bullis[8], Daniella Coker[8], Gabriel Cuellar Partida[8], Devika Dhamija[8], Sayantan Das[8], Sarah L. Elson[8], Teresa Filshtein[8], Kipper Fletez-Brant[8], Pierre Fontanillas[8], Will Freyman[8], Pooja M. Gandhi[8], Karl Heilbron[8], Barry Hicks[8], David A. Hinds[8], Ethan M. Jewett[8], Yunxuan Jiang[8], Katelyn Kukar[8], Keng-Han Lin[8], Maya Lowe[8], Jey C. McCreight[8], Matthew H. McIntyre[8], Steven J. Micheletti[8], Meghan E. Moreno[8], Joanna L. Mountain[8], Priyanka Nandakumar[8], Elizabeth S. Noblin[8], Jared O'Connell[8], Aaron A. Petrakovitz[8], G. David Poznik[8], Morgan Schumacher[8], Anjali J. Shastri[8], Janie F. Shelton[8], Jingchunzi Shi[8], Suyash Shringarpure[8], Vinh Tran[8], Joyce Y. Tung[8], Xin Wang[8], Wei Wan[8], Catherine H. Weldon[8], Peter Wilton[8], Alejandro Hernandez[8], Corinna Wong[8] & Christophe Toukam Tchakouté[8]

