## [Peer Review File · Communications Biology]

Correlations in sleeping patterns and circadian preference between spousesReviewers' comments:

Reviewer #1 (Remarks to the Author):

In this manuscript, Richmond et al. report on their investigation of sleeping patterns and circadian preference between cohabiting spouses. They leveraged two very large datasets, from UK Biobank (47,420 couples) and 23andMe, Inc. (127,035 couples), and applied a very neat study design. First, they reported a weak positive correlation for sleep duration and a weak negative correlation for chronotype between spouses. These correlations were replicated in both UK Biobank and 23andMe. Then, they used accelerometer data in the UK Biobank and observed similar results to the self-report, with the time of diurnal activity showing a correlation.

The authors then extended the Mendelian randomisation design to investigate the causal effects of one individual on another, what they describe as social or indirect genetic effects, and tested 9 sleep traits. They also performed other sensitivity analyses to test the robustness of the MR, and assessed the role of age, time of wear and birth location.

This study is by far the largest of its kind, and represents an interesting and solid contribution to our understanding of how cohabitation influences sleep behaviour. It is very well designed and executed and written clearly, so it is easy for the reader to follow. It will make a great addition to the literature.

I have a few thoughts that the authors might want to consider, either as part of this paper or for their future work:

- How does occupation and socioeconomic status affect the correlations? (e.g., if the couple members are retired or working)
- How do lifestyle variables and substance use (alcohol and tobacco) patterns affect sleep behaviour correlations in spouses?
- How do BMI and chronic disease status affect sleep behaviour in spouses?

Also, the authors should discuss how their analysis are impacted by the fact that the previous GWAS they used to estimate polygenic risk scores had adjusted for sex differences.

Reviewer #2 (Remarks to the Author):

Weak inverse effects for chronotype were found between spouses in the UK Biobank and 23andMe. One explanation could be that they can better handle the 24-h challenge of caring for toddlers. Do you see an association between the chronotype difference between spouses and the number of children?

Sleep in a relationship may be affected by others living in the same household (e.g., pets or children). Furthermore, imposed work schedules can affect the time when people go to sleep. Do these factors mediate or mitigate some of the tested associations?

Do you know whether couples lived in an apartment building or a one-family house? Any differences in how concordantly spouses slept?

Does it matter whether you live in an urban or rural area for how well spouses sleep concordantly?

What about sleep associations among homosexual couples? Similar associations like those observed between heterosexual couples?

Variance in sleep is not only a matter of genes; it highly depends on the state in which a person or a couple is. Thus, I'm curious whether sleep overlaps between spouses may be more or less pronounced

in certain life situations, such as low or high socioeconomic deprivation, taking care of a child, or before and after retirement. Likewise, it may be interesting to see whether couples spending more time together or have similar activity patterns during wake time (which would mean that they may share similar wake experiences resulting in a comparable homeostatic sleep pressure) are more likely to sleep concordantly than those who live their "own" lives.

Reviewer #3 (Remarks to the Author):

Thanks for inviting me for reviewing this paper. Richmond et al. conducted a Mendelian randomization (MR) study exploring correlations in sleeping patterns and circadian preference between spouses, and concluded that individual's sleep may impact that of their partner. Although the paper is well written, this is an obvious observation and the justification that this warrants an MR seems very weak. Public health implications of this study are also not clear and convincing (i.e., providing insights into human sleep interactions). Novelty is also a major concern.

I have some more specific comments, as below:

1. In introduction, the authors are suggested to provide stronger argument why determine the existence and directionality in correlated sleep patterns between spouses matters. Will the results have any possibility in changing the current or future clinical or public health practice. Or by clarifying this research question, can we gain better understanding about etiology?
2. All sleep questions in UK Biobank are very simple and not validated. Also, no information regarding the reliability of these questions has been provided.
3. The interpretation of correlations are not clear. Need to explain why not used beta coefficients, which is more commonly used in MR in this study.
4. Interpretation of SEs in some tables (i.e., Table 2) is difficult because of the lack of units for some variables and reference groups.
5. It's also suggested to provide the R code of the data analysis for reviewers and readers to check.

COMMSBIO-23-0514-T: Response to reviewers

Editor comments:

Your manuscript entitled "Correlations in sleeping patterns and circadian preference between spouses" has now been seen by 3 referees, whose comments are appended below. You will see from their comments copied below that while they find your work of potential interest, they have raised quite substantial concerns that must be addressed. In light of these comments, we cannot accept the manuscript for publication, but would be interested in considering a revised version that addresses these serious concerns.

We hope you will find the referees' comments useful as you decide how to proceed. Should further experimental data or analysis allow you to address these criticisms, we would be happy to look at a substantially revised manuscript. However, please bear in mind that we will be reluctant to approach the referees again in the absence of major revisions.

In particular, we ask that you investigate the impact of other socioeconomic or lifestyle factors as outlined by Reviewers #1-2, and address the contextual and statistical concerns mentioned by Reviewer #3.

Thank you for considering our work for publication in Communications Biology and for permitting us a generous amount of time to revise the manuscript and address the referee comments in full. We have now extended the effect modification analysis to consider other socio-economic and lifestyle factors which might affect the relationships between sleep traits among spouses, including an assessment of employment status, children living in household, type of household, socio-economic position, rural/urban living and daytime activity timing, as suggested by Reviewers #1 and #2. As requested by Reviewer #3, we have also provided more rationale for the study and potential public health implications in the Background and Discussion sections of the manuscript, have clarified the statistical queries and have made the statistical code publicly available via a GitHub repository (https://github.com/rcrichmond/spousal_sleep).

Reviewers' comments:

Reviewer #1 (Remarks to the Author):

In this manuscript, Richmond et al. report on their investigation of sleeping patterns and circadian preference between cohabiting spouses. They leveraged two very large datasets, from UK Biobank (47,420 couples) and 23andMe, Inc. (127,035 couples), and applied a very neat study design. First, they reported a weak positive correlation for sleep duration and a weak negative correlation for chronotype between spouses. These correlations were replicated in both UK Biobank and 23andMe. Then, they used accelerometer data in the UK Biobank and observed similar results to the self-report, with the time of diurnal activity showing a correlation.

The authors then extended the Mendelian randomisation design to investigate the causal effects of one individual on another, what they describe as social or indirect genetic effects, and tested 9 sleep traits. They also performed other sensitivity analyses to test the robustness of the MR, and assessed the role of age, time of wear and birth location.

This study is by far the largest of its kind, and represents an interesting and solid contribution to our

understanding of how cohabitation influences sleep behaviour. It is very well designed and executed and written clearly, so it is easy for the reader to follow. It will make a great addition to the literature.

We thank the reviewer for their positive appraisal of our manuscript.

I have a few thoughts that the authors might want to consider, either as part of this paper or for their future work:

1) How does occupation and socioeconomic status affect the correlations? (e.g., if the couple members are retired or working)

We have extended our sensitivity analyses to assess a number of family-level effect modifiers which may affect the magnitude of the causal relationships between sleep traits among spouses. This includes an assessment of employment status (both employed, one employed and one unemployed/retired, both unemployed/retired) and socioeconomic position (Townsend deprivation index tertiles). Neither employment status nor socioeconomic position appeared to strongly influence the causal effects which were revealed in the main analysis, i.e. chronotype, sleep duration, snoring and L5-timing (Supplementary Figure 8).

We have updated the Methods and Results to incorporate this additional analysis (Methods, pages 23-24, lines 722-747 and Results, page 9, lines 253-280).

2) How do lifestyle variables and substance use (alcohol and tobacco) patterns affect sleep behaviour correlations in spouses? How do BMI and chronic disease status affect sleep behaviour in spouses?

We have included an assessment of how family-level factors may influence the magnitude of the causal effects of sleep traits between spouses, as suggested by Reviewers #1 and #2. This includes an assessment of employment status, children living in household, type of household, socio-economic position, rural/urban living and daytime activity timing (Supplementary Figure 8; Methods, pages 23-24, lines 722-747 and Results, page 9, lines 253-280). How individual-level traits such as alcohol, smoking, BMI and chronic disease status also affect sleep behaviour correlations in spouses is very worthy of investigation. We are currently conducting as part of future work which will also assess these traits in a causal framework to evaluate whether they are causes or consequences of sleep concordance/discordance between spouses.

3) Also, the authors should discuss how their analysis are impacted by the fact that the previous GWAS they used to estimate polygenic risk scores had adjusted for sex differences.

While previous GWAS have adjusted for sex, the polygenic risk scores derived from the GWAS summary statistics strongly associated with sleep traits in both males and females in UK Biobank. This is demonstrated in the assessment of instrument strength shown in Table 3. For some of the accelerometer-derived traits, F-statistics were small which could indicate weak instrument bias. This was particularly the case for accelerometer traits in females, suggesting that there may be some sex differences in the genetic contribution to the sleep traits. However, there would be a necessary trade-off with lower sample sizes for the sex-specific rather than sex-combined GWAS, which may reduce statistical power. This is highlighted in the Discussion (page 15, lines 447-452).

Reviewer #2 (Remarks to the Author):

1) Weak inverse effects for chronotype were found between spouses in the UK Biobank and 23andMe. One explanation could be that they can better handle the 24-h challenge of caring for toddlers. Do

you see an association between the chronotype difference between spouses and the number of children?

We have included an assessment of how family-level factors may influence the magnitude of the causal effects of sleep traits between spouses, as suggested by Reviewers #1 and #2. This includes an assessment of employment status, children living in household, type of household, socio-economic position, rural/urban living and daytime activity timing (Supplementary Figure 8). We did not find evidence to suggest that effects in chronotype were substantially different among those spouses who had children living in the household than those who didn't. However, we did find evidence to suggest that diurnal activity effects (based on accelerometer-derived L5 timing) were stronger in spouses when there were no children in the household (mean difference = 0.90; 0.20, 1.60 SD per SD with no children vs. -0.11; -0.82, 0.62 SD per SD with one or more child in the household; I² = 74%; PHet=0.05).

We have updated the Methods, Results and Discussion to incorporate this additional analysis (Methods, pages 23-24, lines 722-747; Results, page 9, lines 253-280; Discussion, pages 11-12, lines 334-342).

2) Sleep in a relationship may be affected by others living in the same household (e.g., pets or children). Furthermore, imposed work schedules can affect the time when people go to sleep. Do these factors mediate or mitigate some of the tested associations?

Our assessment of how family-level factors may influence the magnitude of the causal effects of sleep traits between spouses now includes an investigation of the number of children living in the household as well as employment status. While we found some evidence to suggest that number of children in the household might influence concordance in diurnal activity between spouses (as described in response to Reviewer #2, comment #1), we did not find that employment status strongly influenced the effect estimates for chronotype, sleep duration, snoring and L5-timing (Supplementary Figure 8). We have updated the Methods and Results to incorporate this additional analysis (Methods, pages 23-24, lines 722-747 and Results, page 9, lines 253-280).

3) Do you know whether couples lived in an apartment building or a one-family house? Any differences in how concordantly spouses slept?

Our assessment of how family-level factors may influence the magnitude of the causal effects of sleep traits between spouses now includes an investigation of the type of household in which the spouses were dwelling during the first UK Biobank assessment. We did not find evidence that type of household (house/bungalow of flat/maisonette/apartment) influenced the effect estimates for chronotype, sleep duration, snoring and L5-timing (Supplementary Figure 8). We have updated the Methods and Results to incorporate this additional analysis (Methods, pages 23-24, lines 722-747; Results, page 9, lines 253-280).

4) Does it matter whether you live in an urban or rural area for how well spouses sleep concordantly?

Our assessment of how family-level factors may influence the magnitude of the causal effects of sleep traits between spouses now includes an assessment of the urban/rural location of the spouses' households at recruitment into the study. We did not find evidence that rural/urban location influenced the effect estimates for chronotype, sleep duration, snoring and L5-timing (Supplementary Figure 8). We have updated the Methods and Results to incorporate this additional analysis (Methods, pages 23-24, lines 722-747 and Results, page 9, lines 253-280).

5) *What about sleep associations among homosexual couples? Similar associations like those observed between heterosexual couples?*

Unfortunately we were unable to assess sleep associations among homosexual couples as the spouse pairs in UK Biobank were determined using a previous approach for which exclusions were made if couples were of the same sex (1). Similarly, in 23andMe, spouses were determined based on parent-offspring trios and so spouses were mother-father pairs. Since our analysis investigated sex differences in sleep patterns, and since differences in sleep concordance between homosexual and heterosexual couples was not the focus of the paper, we would prefer not to investigate this as part of the current manuscript but agree that this would be interesting to evaluate in future studies.

6) *Variance in sleep is not only a matter of genes; it highly depends on the state in which a person or a couple is. Thus, I'm curious whether sleep overlaps between spouses may be more or less pronounced in certain life situations, such as low or high socioeconomic deprivation, taking care of a child, or before and after retirement.*

Our assessment of how family-level factors may influence the magnitude of the causal effects of sleep traits between spouses includes an investigation of employment status, children living in household, type of household, socio-economic position, rural/urban living and daytime activity timing.

No large differences in effects were found by socio-economic, demographic and lifestyle factors, or accelerometer characteristics. This was except for the accelerometer-derived activity timing, where later diurnal activity of one spouse had a larger effect on the other if spouses lived in households without children. In addition, for sleep duration, larger positive effects were observed at older ages, suggesting a convergence in sleep duration between spouses over time.

*Results of this additional analysis are shown in **Supplementary Figure 8**. We have updated the Methods, Results and Discussion to incorporate this additional analysis (**Methods, pages 23-24, lines 722-747; Results, page 9, lines 253-280; Discussion, pages 11-12, lines 334-342**).*

7) *Likewise, it may be interesting to see whether couples spending more time together or have similar activity patterns during wake time (which would mean that they may share similar wake experiences resulting in a comparable homeostatic sleep pressure) are more likely to sleep concordantly than those who live their "own" lives.*

*We have now investigated whether spouses were more likely to sleep concordantly if they had similar activity patterns during wake time. This was done using the accelerometer-derived measure of M10-timing, which is the most active 10 hours of the day. We assessed whether sleep timing between spouses (based on L5-timing) was affected by the difference in daytime activity (based on M10-timing) between spouses, by assessing heterogeneity in the causal estimates between three groups experiencing the lowest, middle and highest levels of difference in activity patterns during wake time, based on M10-timing (**Supplementary Figure 8**). We found no clear evidence that spouses were likely to sleep more concordantly if they had similar activity patterns during the day.*

Reviewer #3 (Remarks to the Author):

Thanks for inviting me for reviewing this paper. Richmond et al. conducted a Mendelian randomization (MR) study exploring correlations in sleeping patterns and circadian preference between spouses, and concluded that individual's sleep may impact that of their partner.

- 1) *Although the paper is well written, this is an obvious observation and the justification that this warrants an MR seems very weak.*

While it may be anticipated that sleep traits within a couple are correlated, until now only small-scale studies have investigated the extent of this interdependence. Within the UK Biobank, we investigated the correlations between nine sleep traits (5 self-report and 4 accelerometer-derived measures), with replication in 23andMe. A number of correlations were found, including some which were not expected e.g. an inverse correlation between chronotype. To further understand the nature of these correlations, we performed Mendelian randomization analysis in order to i) determine whether they represent true causal effects or confounding by shared environmental factors and ii) establish directionality in any causal effects between spouses. MR analysis provided support for effects of an individual's sleep trait on that of their spouse for chronotype, diurnal activity, sleep duration and snoring.

The justification for both the large-scale observational analysis and Mendelian randomization follow-up are emphasised in the following paragraph in the Introduction (page 4, lines 78-85).

“Previous studies using actigraphy measures to investigate spousal sleeping patterns have been limited in terms of sample size. While larger studies have investigated self-reported sleep traits among spouses (2, 3), they may suffer from bias due to individuals' perception and recall of sleeping patterns, which may differ between men and women. Previous observational studies investigating both self-reported and objectively-assessed sleeping patterns between spouses may also be biased by confounding (i.e. by shared socioeconomic and lifestyle factors) and it can be difficult to determine the directionality in correlated sleep patterns between spouses (i.e. the extent to which one spouse influences the sleep patterns of the other, and vice versa).”

- 2) *Public health implications of this study are also not clear and convincing (i.e., providing insights into human sleep interactions).*

In the Introduction, we state that “Within cohabiting couples, it is of interest to investigate the interdependence in sleep patterns since this may exacerbate sleep problems, which could have social, psychological, and physical health implications” (page 3, lines 54-56). We now elaborate on this by stating:

“Establishing whether the sleep habits of a spouse could serve as a risk factor for an individual's own poor sleep would enhance our understanding of the familial impacts on sleep and sleep-related ill health, and would promote opportunities for interventions aimed at the family level.” (page 3, lines 56-58).

Based on the results of our study, we state in the Discussion that “Our findings provide insights into sleep behaviour among cohabiting spouses which may impact on relationship factors and could have further downstream physical and mental health consequences”. We now elaborate on this by stating:

“Our findings provide insights into sleep behaviour among cohabiting spouses and highlight how certain sleep traits can be influenced by the sleep of a person's spouse. This helps us gain a better understanding of the etiology of poor sleep, which may in turn impact on relationship factors and could have further downstream physical and mental health consequences (4-7). Results of this study promote further investigation into the familial impacts of sleep and sleep-related ill health and raise possible opportunities for sleep interventions aimed at the family level. However, the magnitude of sleep effects was small and whether this level of correlation between spouses contributes towards disease risk, as indicated in (8), requires further investigation.” (page 16, lines 489-496).

While the abstract word limit restricts an extensive discussion of the public health implications of our findings, we have now replaced the phrase “providing insights into human sleep interactions” with “promoting opportunities for sleep interventions at the family-level” (page 2, lines 37-38).

3) Novelty is also a major concern.

We disagree that our study is not novel. Existing studies in the sleep field which assessing concordance in sleep behaviours between spouses are scarce and those that exist have typically investigated a very small number of participants (n<100 couples) (9)(10)(11). The present study therefore serves as a novel addition to the sleep literature by conducting an assessment of sleep behaviour in 47,420 spouse pairs in UK Biobank and 127,035 spouse pairs in 23andMe.

Within the Mendelian Randomization field, while previous studies have investigated spousal concordance and indirect genetic effects in relation to a range of socio-economic, lifestyle and behavioural phenotypes (12, 13), this has not included an assessment of sleep traits. Furthermore, while the majority of these studies have identified positive correlations in traits between spouses (representative of assortative mating, social homogamy or interactions after partnership), the finding of an inverse correlation (and confirmed causal effect) in chronotype between spouses is novel and of interest, since it goes against the plethora of evidence indicating widespread similarities (rather than differences) of spouses for many phenotypes.

I have some more specific comments, as below:

4) In introduction, the authors are suggested to provide stronger argument why determine the existence and directionality in correlated sleep patterns between spouses matters. Will the results have any possibility in changing the current or future clinical or public health practice. Or by clarifying this research question, can we gain better understanding about etiology?

We thank the reviewer for this suggestion. As mentioned in relation to the reviewer’s comment #2, in the Introduction, we now emphasise the potential importance of our research question with the following statement:

“Establishing whether the sleep habits of a spouse could serve as a risk factor for an individual’s own poor sleep would enhance our understanding of the familial impacts on sleep and sleep-related ill health, and would promote opportunities for interventions aimed at the family level.” (page 3, lines 56-58).

5) All sleep questions in UK Biobank are very simple and not validated. Also, no information regarding the reliability of these questions has been provided.

The reviewer raises an important point about the validity of the sleep questionnaire measures which we have now addressed in the Limitations section of the Discussion (page 14, lines 411-420):

“Self-reported measures of the sleep traits in UK Biobank and 23andMe were based on a limited number of questions which may crudely assess underlying sleep traits and/or be subject to bias. However, there is a trade-off between the number of individuals with sleep phenotypes obtained from easy-to-administer questionnaires and more objective or clinically-derived sleep measures, which are typically available on a smaller number of participants. The replication of genetic associations identified in relation to the self-reported sleep traits in UK Biobank with comparable phenotypes in independent studies including 23andMe (14-16), and with objectives measures i.e. from the accelerometer assessment in UK Biobank (16, 17), serves as an indirect means of validation

of the self-report measures (18). Further to this, we also directly assessed spousal concordance using accelerometer-derived measures in a subset of the UK Biobank.”

- 6) *The interpretation of correlations are not clear. Need to explain why not used beta coefficients, which is more commonly used in MR in this study.*

Effect estimates from the MR analysis represent the mean difference in the spouse’s sleep trait (in SD) per SD increase in an individual’s own sleep trait, with the exception of snoring for which estimates represent a risk difference (since this is a binary trait). This interpretation is highlighted in Figure 4 and also in the Results. Corresponding mean differences from both multivariable and MR analysis are directly comparable to phenotypic partial correlations when both exposure and outcome are measured in SD units. This is highlighted in the Methods, “Standardized variables are presented..to allow for direct comparisons with the correlation coefficients estimated for the phenotypic correlation.” (page 23, lines 698-699).

- 7) *Interpretation of SEs in some tables (i.e., Table 2) is difficult because of the lack of units for some variables and reference groups.*

*As above, effect estimates (and corresponding SEs) represent the mean difference in the spouse’s sleep trait (in SD) per SD increase in an individual’s own sleep trait, with the exception of snoring for which estimates represent risk difference. This is described in the **Table 2** footnote.*

- 8) *It’s also suggested to provide the R code of the data analysis for reviewers and readers to check.*

The Stata code used to estimate phenotypic and genetic correlations, and to perform the main one-sample Mendelian randomization analysis in UK Biobank, is available via GitHub (https://github.com/rcrichmond/spousal_sleep).

References

1. Howe LJ, Lawson DJ, Davies NM, St Pourcain B, Lewis SJ, Davey Smith G, et al. Genetic evidence for assortative mating on alcohol consumption in the UK Biobank. *Nat Commun.* 2019;10(1):5039.
2. Arber S, Hislop J, Bote M, Meadows R. Gender Roles and Women's Sleep in Mid and Later Life: A Quantitative Approach. *Sociological Research Online.* 2007;12(5):182-99.
3. Ulfberg J, Carter N, Talback M, Edling C. Adverse health effects among women living with heavy snorers. *Health Care Women Int.* 2000;21(2):81-90.
4. Sofi F, Cesari F, Casini A, Macchi C, Abbate R, Gensini GF. Insomnia and risk of cardiovascular disease: a meta-analysis. *Eur J Prev Cardiol.* 2014;21(1):57-64.
5. Cappuccio FP, D'Elia L, Strazzullo P, Miller MA. Quantity and quality of sleep and incidence of type 2 diabetes: a systematic review and meta-analysis. *Diabetes Care.* 2010;33(2):414-20.
6. Baglioni C, Battagliese G, Feige B, Spiegelhalder K, Nissen C, Voderholzer U, et al. Insomnia as a predictor of depression: a meta-analytic evaluation of longitudinal epidemiological studies. *J Affect Disord.* 2011;135(1-3):10-9.
7. Haus EL, Smolensky MH. Shift work and cancer risk: potential mechanistic roles of circadian disruption, light at night, and sleep deprivation. *Sleep Med Rev.* 2013;17(4):273-84.
8. Gunn HE, Buysse DJ, Matthews KA, Kline CE, Cribbet MR, Troxel WM. Sleep-Wake Concordance in Couples Is Inversely Associated With Cardiovascular Disease Risk Markers. *Sleep.* 2017;40(1).
9. Pankhurst FP, Horne JA. The influence of bed partners on movement during sleep. *Sleep.* 1994;17(4):308-15.
10. Meadows R, Arber S, Venn S, Hislop J, Stanley N. Exploring the interdependence of couples' rest-wake cycles: an actigraphic study. *Chronobiol Int.* 2009;26(1):80-92.
11. Randler C, Kretz S. Assortative mating in morningness-eveningness. *Int J Psychol.* 2011;46(2):91-6.
12. Xia C, Canela-Xandri O, Rawlik K, Tenesa A. Evidence of horizontal indirect genetic effects in humans. *Nat Hum Behav.* 2021;5(3):399-406.
13. Robinson MR, Kleinman A, Graff M, Vinkhuyzen AAE, Couper D, Miller MB, et al. Genetic evidence of assortative mating in humans. *Nature Human Behaviour.* 2017;1(1):0016.
14. Jones SE, Lane JM, Wood AR, van Hees VT, Tyrrell J, Beaumont RN, et al. Genome-wide association analyses of chronotype in 697,828 individuals provides insights into circadian rhythms. *Nat Commun.* 2019;10(1):343.
15. Jansen PR, Watanabe K, Stringer S, Skene N, Bryois J, Hammerschlag AR, et al. Genome-wide analysis of insomnia in 1,331,010 individuals identifies new risk loci and functional pathways. *Nat Genet.* 2019;51(3):394-403.
16. Dashti HS, Jones SE, Wood AR, Lane JM, van Hees VT, Wang H, et al. Genome-wide association study identifies genetic loci for self-reported habitual sleep duration supported by accelerometer-derived estimates. *Nat Commun.* 2019;10(1):1100.
17. Jones SE, van Hees VT, Mazzotti DR, Marques-Vidal P, Sabia S, van der Spek A, et al. Genetic studies of accelerometer-based sleep measures yield new insights into human sleep behaviour. *Nat Commun.* 2019;10(1):1585.
18. Lane JM, Qian J, Mignot E, Redline S, Scheer F, Saxena R. Genetics of circadian rhythms and sleep in human health and disease. *Nat Rev Genet.* 2023;24(1):4-20.

REVIEWERS' COMMENTS:

Reviewer #1 (Remarks to the Author):

The authors have addressed my concerns to a reasonable degree, and I believe that the revised manuscript is acceptable for publication. Again, I congratulate the authors for doing a neat and insightful study.

Reviewer #2 (Remarks to the Author):

You have satisfactorily addressed my comments and concerns.

Reviewer #3 (Remarks to the Author):

The authors have addressed my questions and comments satisfactorily. I have no further comments.